# Preparation and Characterization of Mesocarbon Microbeads by the Co-Polycondensation of High-Temperature Coal Tar Pitch and Coal Pyrolytic Extracts

**DOI:** 10.3390/ma15155136

**Published:** 2022-07-24

**Authors:** Lidong Yan, Yilin Fang, Jianfeng Deng, Yaming Zhu, Yuzhu Zhang, Junxia Cheng, Xuefei Zhao

**Affiliations:** 1Institute of Chemical Engineering, University of Science and Technology Liaoning, Anshan 114051, China; cctv3392@163.com (L.Y.); a1044520381@163.com (Y.F.); a13024900214@163.com (J.D.); zhangyz105@163.com (Y.Z.); cheng.anshan@163.com (J.C.); 2Key Laboratory of Chemical Metallurgy Liaoning Province, University of Science and Technology Liaoning, Anshan 114051, China

**Keywords:** high-temperature coal tar pitch, coal pyrolytic extracts, mesocarbon microbeads (MCMB), carbon crystalline

## Abstract

Mesocarbon microbeads (MCMBs) are a kind of engineering and functional artificial carbon materials generally prepared by the polymerization of polycyclic aromatic hydrocarbons. The physicochemical property of the raw materials plays a key role in the quality of MCMBs. For a detailed analysis of the synergistic effects of the generation of MCMBs, a high-temperature coal tar pitch was used as raw materials, and coal pyrolytic extracts were used as additive to synthesize the MCMBs. The microstructure and morphology of the derived MCMBs were determined by an optical microscope, scanning electron microscope, X-ray diffraction, Raman spectrum, and laser particle size analyzer. In fact, the addition of the coal pyrolytic extracts can adjust the molecular structure of the blending pitch, and the coal pyrolytic extracts can promote the generation of the MCMBs during the co-polycondensation process. The MCMBs obtained by co-polycondensation method have a good degree of sphericity, lower defects in the surface morphology, and a lower charge transfer resistance (R_ct_) of 4.677 Ω.

## 1. Introduction

Mesocarbon microbeads (MCMBs) are recognized as a kind of soft carbon with good sphericity, uniform particle size distribution, ease of graphitization, and other special properties [1]. MCMBs are accepted as a suitable precursor widely applied in a lithium-ion battery [1,2], lithium-ion supercapacitor [3], Li-S battery [4], sodium-ion battery [5], MCMB–SiC composite ceramic [6], solid acid catalyst [7], photocatalyst [8], and so on. In other words, MCMBs are one of the most important functional artificial carbon materials widely used in the metallurgy industry, chemical industry, aerospace industry, energy storage, and catalysis [9].

Generally, MCMBs are prepared by the thermal polycondensation of aromatic compounds, including coal tar pitch, petroleum pitch, heavy oil, anthracene oil, naphthalene, and biomass tar [10,11,12,13,14]. What is more, some researchers also report using a supercritical fluid extraction method and suspension method to prepare MCMBs with mesophase pitch as raw materials [15]. So far, a thermal polycondensation method is a common, economical, and friendly way to produce MCMBs. However, MCMBs produced by thermal polycondensation with a coal tar pitch as raw materials present some defects; for example, the particle size is hard to control, the yield is low, and the internal microstructure and the morphology of the surface are always not easy to control.

Actually, the particle size distribution, surface roughness, and internal microstructure of MCMBs have a serious impact on their properties. As reported in the literature [16,17], a smaller diameter of MCMBs always means a higher capacity of a lithium-ion battery with the MCMBs as anode materials. Researchers choose the co-polycondensation method to improve the properties of MCMBs. The co-polycondensation method can be divided into two aspects according to the type of additives. Briefly, some researchers [18,19,20] have studied the influence of inorganic additives (ZnO, TiO_2_, and SiO_2_) on the electrochemistry property of derived MCMBs by the co-polycondensation of a coal tar pitch and inorganic additives. Li et al. [12] and Yan et al. [21] studied the effects of organic additives (biomass tar pitch and direct coal liquefaction pitch) on the yield of MCMBs by the co-polycondensation of a coal tar pitch and organic additives. In other words, co-polycondensation is one of the useful methods to enhance the property of the derived MCMBs.

Coal pyrolytic extracts (CPEs) are a kind of aromatic hydrocarbons obtained from low-rank coal (such as long-flame coal and brown coal). Theoretically, CPEs have a higher reaction reactivity to generate the mesophase [11,12]. In order to determine the synergistic effect (influence on the particle distribution, surface morphology, carbon microcrystalline, and resistivity) on the production of MCMBs by co-polycondensation of a coal tar pitch and CPEs, MCMBs were prepared by the thermal polycondensation of a coal tar pitch and CPEs at varied appending proportions in this paper.

## 2. Experimental Sections

### 2.1. Raw Materials

A high-temperature coal tar pitch (HCTP) with a softening point (SP) of 81 °C, toluene insoluble (TI) of 18.23%, quinoline insoluble (QI) of 2.10%, and coking value (CV) of 51.75% was obtained from Angang Steel Co. Ltd. (Anshan, China). Coal pyrolytic extracts (CPEs) with an SP of 90 °C, TI of 23.92%, QI of 0.89%, and CV of 45.09% were pyrolytically extracted by washing oil from brown coal at a temperature of 400 °C. Quinoline (AR) and toluene (AR) were purchased from Shenyang Chemical Industry (Shenyang, China).

### 2.2. Preparation of MCMB

An amount of 150 g of pitch mixture (blending of HCTP and CPE at a varied ratio) was put into an atmospheric stainless steel reactor. The modified pitch was obtained under certain conditions (reaction temperature of 430 °C, holding time of 4.5 h, and N_2_ as inner gas). Green MCMBs were obtained by solvent extraction with washing oil as the solvent and toluene as the cleaning agent. The MCMBs were finally taken from the calcination of the green MCMBs. Briefly, the calcination temperature was 1300 °C, and pure N_2_ was used as the protecting gas during the calcination process. The preparation process of MCMBs is presented in Figure 1. The green MCMBs and calcinated MCMBs are marked as MCMB-g-X and MCMB-X, respectively. Factually, X means the content of CPE in the blending pitch. For example, MCMB-1.5% means that the MCMBs were prepared by co-polycondensation of the blending pitch with the content of CPE at 1.5%.

### 2.3. Characterization 

Proximate analysis (including SP, TI, QI, and CV) was tested according to national standards of the People’s Republic of China, reported in the related literature [22]. The element content was determined on an elemental analyzer (Vario ELIII, Elementar, Germany). The functional group was carried out on a spectrometer (Nicolet iS5 FTIR, Thermo Nicolet Corporation, Madison, USA). The thermal stability was tested in a TGA analyzer (TAQ500, TA, USA). The distribution of particle size was tested in a laser particle size analyzer (MS2000, Malvern Panalytical, USA).

The optical microstructure was tested in a polarizing microscope (Axio Scope A1 Pol, Carl Zeiss, Jena, Germany). The surface morphology was carried out on a scanning electron microscope (IGMA-HD Carl Zeiss, Germany). The distribution of carbon crystalline was determined by Raman spectra analysis (LabRAM HR Evolution, Jobin Yvon, Longjumeau, France) and X-ray diffraction (PANalytical B.V., Almelo, The Netherlands), respectively.

## 3. Results and Discussion

### 3.1. Ultimate Analysis of HCTP and CPE

Ultimate analysis was one of the most important indexes to reflect the basic physicochemical properties of complex organic compounds. The ultimate analysis of HCTP and CPE is shown in Table 1.

As presented in Table 1, the content of C in HCTP was higher than that in CPE, but the content of H in HCTP was lower than that in CPE. As a result, the C/H ratio in HCTP was 1.81, which was much higher than that in CPE. What is more, the oxygen content in HCTP was 4.57%, but the oxygen content in CPE was 9.31%. In other words, there were more oxygen-containing functional groups in CPE molecules than in HCTP.

### 3.2. FTIR Analysis of HCTP and CPE

FTIR was accepted as an essential method to characterize the molecular structure of coal pitch and coal-derived materials. An FTIR graph of HCTP and CPE is shown in Figure 2.

As shown in Figure 2, the absorption peaks of the HCTP and CPE were extremely similar. However, the intensities of some functional groups were significantly varied in these two samples. The major differences of the functional groups of HCTP and CPE were -CH_2_- (the absorption peaks at wavenumbers of 2923 and 2853 cm^−1^ were caused by the asymmetric -CH_2_- and symmetric -CH_2_-, respectively [23]) and C=O (which is marked out in Figure 2). Briefly, the transition intensities of -CH_2_- and C=O in CPE were much higher than those in HCTP. In other words, there were more -CH_2_- and C=O groups in CPE. CPE was obtained by pyrolysis extracted by washing oil from brown coal at a temperature of 400 °C, so the aromatic condensation index was lower than HCTP (the byproduct of the coking industry, which reacted at 1000 °C), but the content of O was much higher than HCTP. This result was identical with the result of ultimate analysis.

### 3.3. Polarized Microstructure of Green MCMBs

The polarized microstructure of MCMBs is usually characterized by an optical microscope. The polarized microstructure of green MCMBs from the co-polycondensation of HCTP and CPE at varied ratios is presented in Figure 3.

As shown in Figure 3, there was obviously an “extinction phenomenon” of spheres on each cross section of the modified pitch (which was rich in green MCMBs). In other words, the MCMBs were generally created in the co-polycondensation of HCTP and CPE. What is more, with the increase in the ratio of CPE, the MCMBs were extremely increased in the sample. Briefly, the quantity and the diameter of MCMB-3% were higher than those of MCMB-g-0% and MCMB-g-1.5%. As shown in Figure 3d, there was a “coalesce phenomenon” presented in MCMB-g-5%, so there were more fused MCMBs and large-scale MCMBs in MCMB-g-5%. Actually, there were more irregular mesophases (such as oven-shaped mesophase and peanut-shaped mesophase) generated in MCMB-g-7%, which were attributed to the “coalesce phenomenon” of MCMBs. In fact, the CPE molecules contain more C=O functional groups, which can act as the reactive site during the co-polycondensation of HCTP and CPE. As a result, the MCMBs were more easily generated and grown up during the co-polycondensation of HCTP and CPE. Additionally, the appropriate addition ratio of CPE can generate more MCMBs and size, but the excessive addition of CPE may cause a more irregular mesophase. Thus, the addition of CPE at a ratio of 3% in HCTP to synthesize MCMBs by co-polycondensation method was a perfect content. 

### 3.4. Particle Size of MCMBs

Particle size distribution was one of the most important indexes to judge the quality of MCMBs. The particle size distribution of MCMBs is shown in Figure 4.

The particle size distribution of MCMBs is presented Figure 4, and the particle size distribution of each MCMBs is presented in a normal distribution form. The particle size of each MCMB was in the order of MCMB-0% < MCMB-1.5% < MCMB-3% < MCMB-5% < MCMB-7%. The parameters of particle size of varied MCMBs are listed in Table 2.

As listed in Table 2, both the median particle size (*D*50) and 90% content particle size (*D*90) of MCMB-0%, MCMB-1.5%, MCMB-3%, MCMB-5%, and MCMB-7% were gradually increased. MCMB-0% had the lowest volume average particle size (D(4, 3)) of 16.78 μm, and MCMB-7% had the highest D(4, 3) of 29.97 μm. Generally, the uniformity of the particle size affects the quality of the MCMBs. In order to determine the uniformity of the particle size directly, a uniformity index of particle size (*U*) was marked. Actually, the calculated method of *U* is shown in Equation (1).
(1)U=(D90−D50)D50

In fact, the higher *U* also meant good uniformity of the particle size. As listed in Table 2, MCMB-3% had the highest *U* of 0.72, which was higher than other MCMBs. This result was matched with the result of the polarized microstructure of MCMBs (shown in Figure 3). In other words, the additive of CPE at a content of 3% in HCTP was the optimum proportion to generate high-quality MCMBs.

### 3.5. Surface Morphology of MCMBs

A scanning electron microscope (SEM) is generally used as a useful method to characterize the surface morphology of materials. SEM images of MCMBs are shown in Figure 5.

As presented in Figure 5, MCMB-0%, MCMB-1.5%, and MCMB-3% were shaped as a regular sphere pattern, which were the standard MCMBs. There was more floc particulate matter covered on the surface of MCMB-0%; this phenomenon is attributed to the primary quinoline insoluble (QI) on HCTP [10]. As reported in the literature [17], the primary QI in the raw coal pitch may be covered on the surface of the MCMBs during the generation process by polycondensation method. With the increase in the additive rate of CPE, the floc particulate matter on the surface was significantly decreased. What is more, there were extremely no pore or other defects on the surface of MCMB-3%. As presented in Figure 5e,g, the surfaces of MCMB-5% and MCMB-7% were smooth but had more big pores. Well, MCMB-5% and MCMB-7% were shaped as an irregular sphere pattern. Much of MCMBs in MCMB-5% and MCMB-7% were peanut-shaped mesophase and silkworm cocoon-shaped mesophase. In fact, with the improvement of the addition of CPE and the content of primary QI in the mixed pitches were decreased clearly. As a result, the surface of MCMBs prepared by co-polycondensation of HCTP and CPE was smoother than that prepared by HCTP. On the other hand, there were more C=O and R_2_CH_2_- functional groups in CPE; these functional groups can act as a reaction site to promote the reaction rate of polycondensation. Therefore, there were more big pores on the surface of MCMB-5% and MCMB-7%.

### 3.6. XRD Analysis of MCMBs

The combination of XRD and a curve-fitted method was used to characterize the carbon microcrystalline of carbon materials. XRD graphs and a curve-fitted graph of MCMBs are shown in Figure 6.

As shown in Figure 6a, the position of diffraction peaks in varied MCMB samples was the same (at the position near 26°) and is called as the typical characteristic peaks of (002) of carbon/graphite materials. The shape and position of C(002) in the five MCMB samples were similar, but the intensity was varied. As reported in the literature [24], the C(002) of nongraphitized carbon consisted with γ-band and π-band around 20° and 26°, respectively. In order to characterize the microcrystalline of MCMBs, the content of trended standardized carbon microcrystalline (*I_g_*) was used to predict the graphitization characteristics. Actually, the equation of *I_g_* is presented as Equation (2):(2)Ig=Aπ(Aπ+Aγ)×100%
where *Aγ* and *Aπ* mean the curve-fitted area of γ-band and π-band, respectively. The *I_g_*’s of MCMBs are presented in Table 3.

As shown in Table 3, the I_g_’s of MCMB-0%, MCMB-1.5%, MCMB-3%, MCMB-5%, and MCMB-7% were 82.49%, 81.07%, 83.44%, 79.57%, and 79.07%, respectively. In other words, MCMB-3% had the highest content of trended standardized carbon microcrystalline. This phenomenon can be attributed to the microstructure of MCMBs. MCMB-3% had the minimum defects and pore, so the molecular structure in the green MCMBs was easier to aromatize during the calcination process.

### 3.7. Raman Analysis of MCMBs

Raman spectrum combined with a curve-fitted method was another useful method to judge the carbon microcrystalline of carbon materials. A graph of a Raman spectrum and a curve-fitted graph of MCMBs are shown in Figure 7.

As presented in Figure 7, there were two significant peaks near the Raman shifts of 1360 and 1580 cm^−1^, bounded to D band and G band, respectively. As shown in Figure 7a, the position and the shape of these two peaks in each MCMB sample were extremely similar, but the intensity and the width were varied. According to the literature [24], the asymmetrically broad peak was superimposed by the diffraction curves of different carbon microcrystallines in MCMBs. Actually, the Raman spectrum graph of the MCMBs was divided into five curves marked as D1 peak, D2 peak, D3 peak, D4 peak, and G peak [25]. The distribution of the carbon microcrystalline in MCMBs is listed in Table 4.

As listed in Table 4, the contents of graphite microcrystalline in MCMB-0%, MCMB-1.5%, MCMB-3%, MCMB-5%, and MCMB-7% were 12.31%, 12.64%, 13.95%, 13.51%, and 13.49%, respectively. In other words, MCMB-3% had the highest content of graphite microcrystalline at 13.95%. However, MCMB-3% had the lowest content of amorphous carbon at 16.75%. In fact, the higher content of graphite microcrystalline and the lower content of amorphous carbon always meant good graphitization characteristics. As presented in Figure 3 and Figure 5, MCMB-3% had better optical anisotropy and morphology and also lower defects than other MCMB samples. As a result, the carbon microcrystalline in MCMB-3% was more neat and easier to graphitize. Briefly, the distribution of carbon microcrystalline determined by the Raman spectrum was consistent with the results calculated by XRD.

### 3.8. Alternating Current Impedance Spectrum Analysis of MCMBs

It was generally accepted that EIS was the desired method to determine the electronic transmission performance of carbon materials. An EIS graph of MCMBs is shown in Figure 8.

All of the five kinds of MCMBs had a basically similar type of EIS and consisted of two regions. Briefly, the approximate straight line was named low-frequency region, and the approximate semicircle zone was named high-frequency region. The radius of curvature reflects the velocity of electrons from the surface of the material. Actually, ZSimpWin software was used to curve-fit the EIS of MCMBs, and the curve-fitted results are listed in Table 5.

As shown in Table 5, the charge transfer resistances (R_ct_) of MCMB-0%, MCMB-1.5%, MCMB-3%, MCMB-5%, and MCMB-7% were 7.555 Ω, 4.941 Ω, 4.677 Ω, 4.981 Ω, and 5.988 Ω, respectively. Actually, a lower R_ct_ means a higher electron activity of carbon materials. In other words, MCMB-3% has a higher electrical property. The order of electrical conductivity of these five MCMBs was: MCMB-3% > MCMB-1.5% > MCMB-5% > MCMB-7% > MCMB-0%. This phenomenon can be illustrated by the content of graphite microcrystalline and pore distribution in the MCMBs. Higher contents of graphite microcrystalline and lower pore distribution in carbon materials always mean good electrical properties [26].

## 4. Conclusions

The five kinds of MCMBs were obtained by co-polycondensation of a high-temperature coal tar pitch (HCTP) and coal pyrolytic extracts (CPEs). The addition of CPEs had great effects on the reaction activity of the blending pitch during the co-polycondensation process. The microstructure (including optical microstructure and carbon microcrystalline), surface morphology, and resistance of the derived MCMBs were determined by a polarized microscope, XRD and Raman spectrum, and EIS, respectively. The co-polycondensation of HCTP and CPEs (mixture of HCTP and CPEs at a suitable ratio) was a good method to produce high-quality MCMBs. In fact, the suitable ratio of CPEs as additives can improve the content of graphite microcrystalline and decrease the content of amorphous carbon and resistance in MCMBs with the co-polycondensation of HCTP and CPEs. CPEs were good additives to adjust the molecular distribution and reaction reactivity of HCTP to generate MCMBs. The co-polycondensation of HCTP and a moderate ratio of CPEs was a good method to generate high-quality (good degree of sphericity, lower defects in the surface morphology, and lower charge transfer resistance) MCMBs.

## Figures and Tables

**Figure 1 materials-15-05136-f001:**
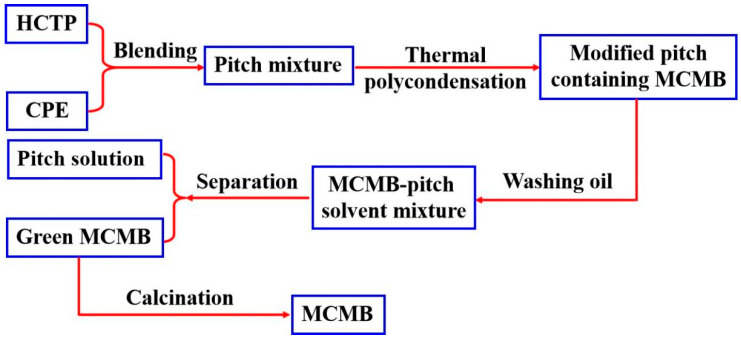
The preparation process of MCMBs.

**Figure 2 materials-15-05136-f002:**
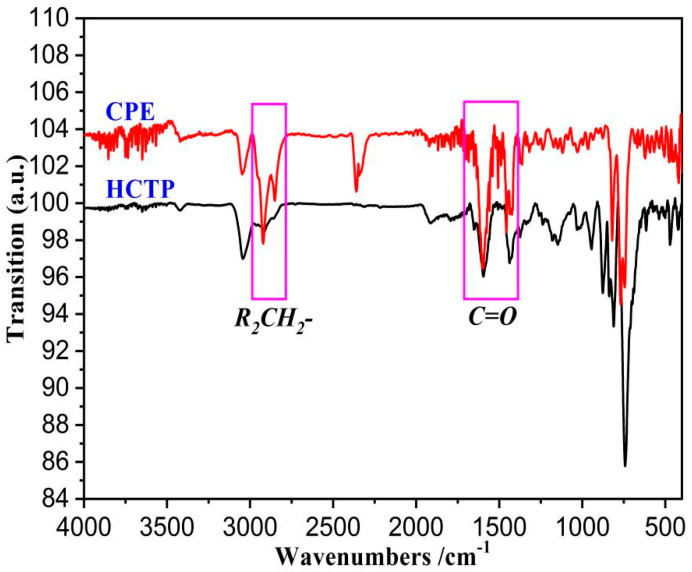
The FTIR spectrum of raw materials.

**Figure 3 materials-15-05136-f003:**
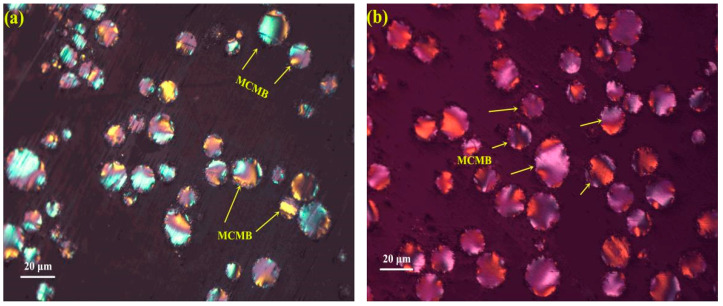
Optical microstructure of MCMBs: (**a**) MCMB-g-0%, (**b**) MCMB-g-1.5%, (**c**) MCMB-g-3%, (**d**) MCMB-g-5%, and (**e**) MCMB-g-7%.

**Figure 4 materials-15-05136-f004:**
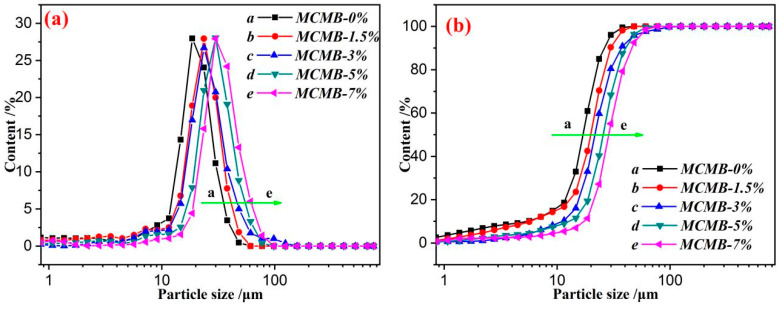
Particle size distribution of MCMBs: (**a**) differential distribution curves and (**b**) cumulative distribution curves.

**Figure 5 materials-15-05136-f005:**
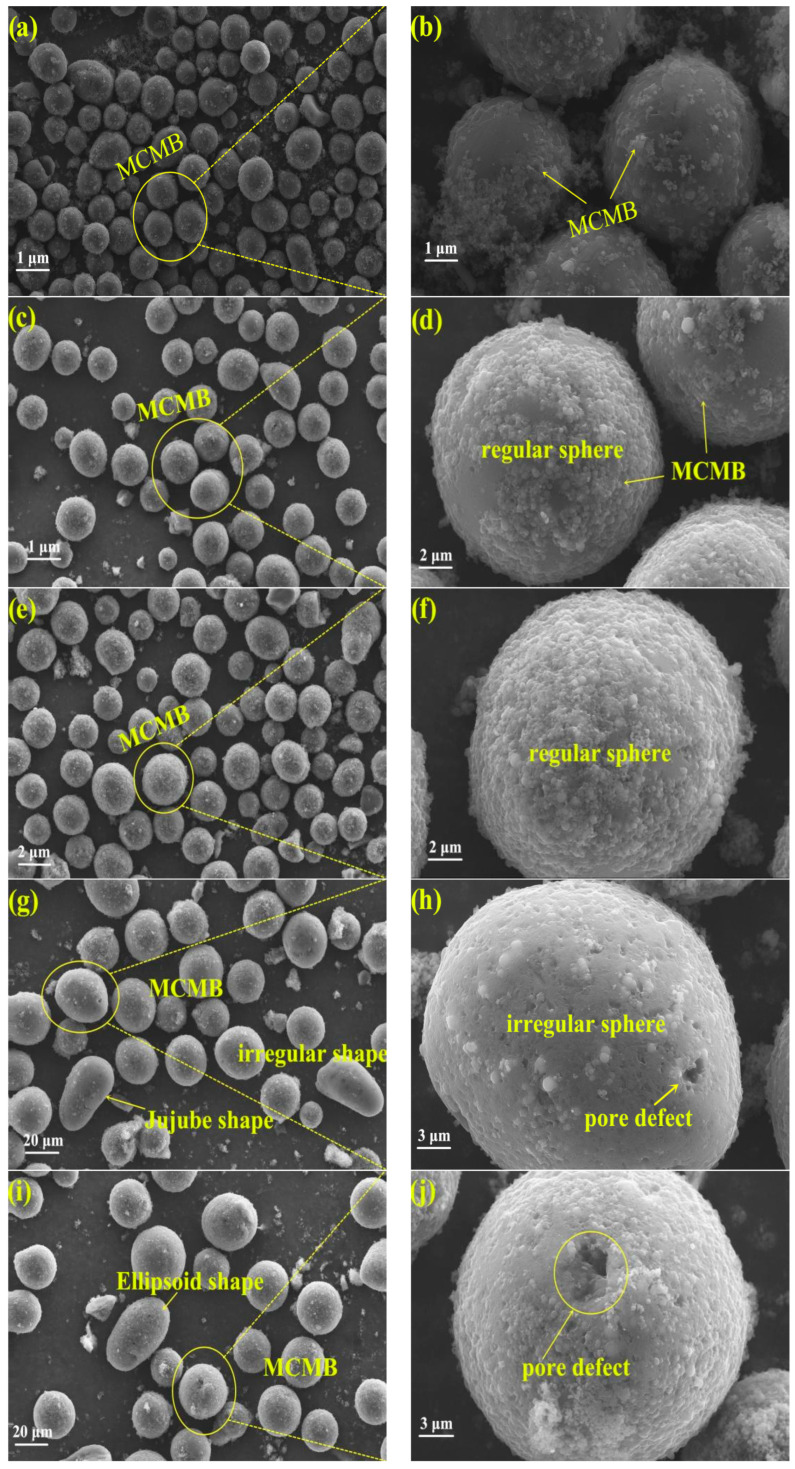
The surface morphology of MCMBs for: (**a**,**b**) MCMB-0%, (**c**,**d**) MCMB-1.5%, (**e**,**f**) MCMB-3%, (**g**,**h**) MCMB-5%, (**i**,**j**) for MCMB-7%.

**Figure 6 materials-15-05136-f006:**
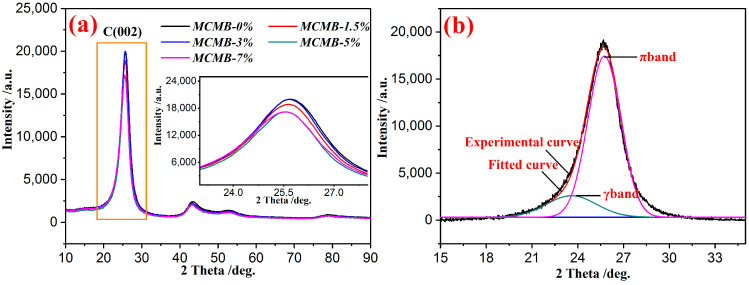
XRD graphs of MCMBs (**a**) and a curve-fitted graph of MCMB-3% (**b**).

**Figure 7 materials-15-05136-f007:**
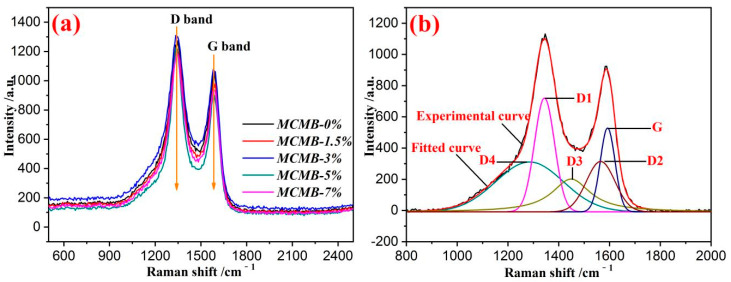
Raman spectrum of MCMBs (**a**) and fitted curve of MCMB-3% (**b**).

**Figure 8 materials-15-05136-f008:**
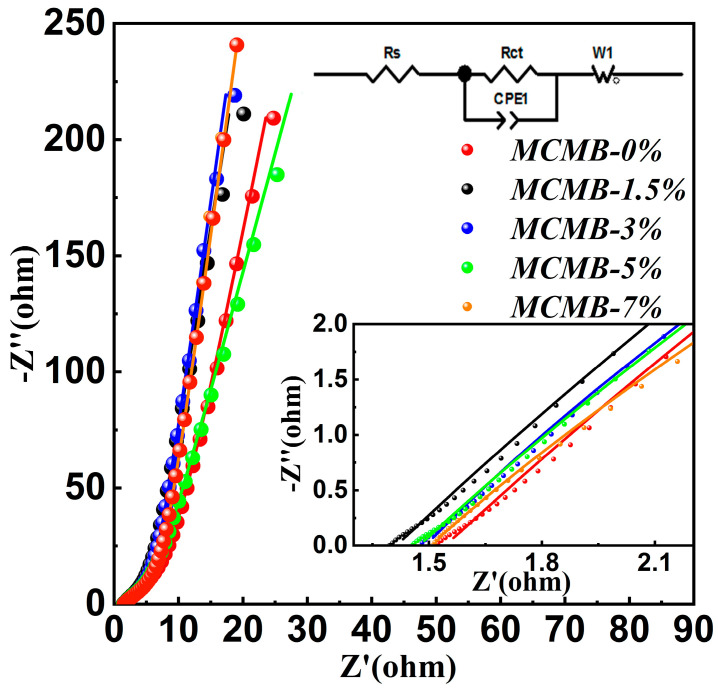
Alternating current impedance spectrum of MCMBs.

**Table 1 materials-15-05136-t001:** Ultimate analysis of HCTP and CPE.

Samples	C/%	H/%	N/%	S/%	O*/%	C/H
HCTP	89.56	4.12	1.14	0.61	4.57	1.81
CPE	84.18	4.53	1.57	0.41	9.31	1.55

Note: O* means the content of oxygen was determined by subtraction method.

**Table 2 materials-15-05136-t002:** The parameters of particle size in MCMBs.

Samples	D50 (μm)	D90 (μm)	D(4, 3) (μm)	U*
MCMB-0%	16.96	26.28	16.78	0.55
MCMB-1.5%	19.83	29.78	19.41	0.50
MCMB-3%	21.63	37.20	23.86	0.72
MCMB-5%	25.60	40.57	26.20	0.58
MCMB-7%	28.64	45.93	29.97	0.60

U* means the uniformity index of particle size.

**Table 3 materials-15-05136-t003:** The I_g_’s of MCMBs.

Samples	γ	π	Aγ	Aπ	Ig/%
MCMB-0%	23.57565	25.67931	6406.26182	30,171.5896	82.49
MCMB-1.5%	23.7749	25.72322	7370.29101	31,561.16705	81.07
MCMB-3%	23.5886	25.74459	6543.85787	32,977.80438	83.44
MCMB-5%	23.48543	25.61029	6742.52079	26,262.90231	79.57
MCMB-7%	23.64086	25.5899	7292.6567	27,555.09149	79.07

**Table 4 materials-15-05136-t004:** The distribution of carbon microcrystalline in MCMBs.

Samples	I_D1_	I_D2_	I_D3_	I_D4_	I_G_	I_G_/I_All_	I_D3_/I_All_
MCMB-0%	48,307.03	30,446.43	36,422.87	71,154.28	26,150.28	12.31	17.14
MCMB-1.5%	48,569.87	30,117.34	38,907.44	79,668.92	28,547.29	12.64	17.23
MCMB-3%	48,760.46	25,364.84	33,454.24	64,318.97	27,863.58	13.95	16.75
MCMB-5%	38,189.04	22,729.50	34,637.22	56,431.36	23,735.38	13.51	19.71
MCMB-7%	42,814.085	28,622.92	33,024.65	60,251.85	25,688.22	13.49	17.34

**Table 5 materials-15-05136-t005:** Resistance of MCMBs calculated from an alternating current impedance spectrum.

Samples	R_s_ (Ω)	R_ct_ (Ω)	W (Ω)/10^−5^
MCMB-0%	1.540	7.555	11.032
MCMB-1.5%	1.489	4.941	10.952
MCMB-3%	1.410	4.677	10.764
MCMB-5%	1.472	4.981	12.840
MCMB-7%	1.500	5.988	9.459

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
