# Peer review of "Preparation and Characterization of Mesocarbon Microbeads by the Co-Polycondensation of High-Temperature Coal Tar Pitch and Coal Pyrolytic Extracts"

_materials, 2022, doi:10.3390/ma15155136_

Round 1
Reviewer 1 Report
The authors present a well-structured work about Mesocarbon Microbeads derivatives and its characterization.
There are few issues that must be addresed:
1. References are not presented applying the journal isntructions for authors (https://www.mdpi.com/journal/materials/instructions).
2. The degree Celsius symbol looks strange. Is it modified?
3. What information can the authors obtain from the 2250-2500 cm-1 wavenumber range of the CPE FTIR plot?
4. Fig.4 b): x-axis units should be checked.
5. The use of Equation 1 does not obtain the data presented in Table 2.
6. Figure 5 caption: there are two e)-f) sets, please modify it.
7. Use "Fig." or "Figure" for figures captions, but not both.
Author Response
We thank the reviewer 1 for the decent review comments that lead to improvements of the revised manuscript.Please see below for responses and revisions made based on the reviewer’s comments.

Reviewer 2 Report
The manuscript suggested by the authors is interesting, easy to follow and well written. Brings useful and new knowledge to experts in this field. I suggest that the paper be accepted in this form with the only change in Section 4. Conclusion: to give an overview of the most significant results obtained in the paper.
Author Response
We thank the reviewer 2 for the decent review comments that lead to improvements of the revised manuscript.Please see below for responses and revisions made based on the reviewer’s comments.

Reviewer 3 Report
1- This is an interesting work of Yan and coworkers on a topic that is recurrent in the field. I am fine with the content of the manuscript and the results presented seem to be reasonable from scientific point of view. However, there are several grammatical errors all over the ms, starting from the abstract of the ms itself. One such example could be “ The optical microscope, scanning electron microscope, X-ray diffraction, Raman spectrum, and laser particle size analyzer has been used to determine the microstructure and morphology of the derived MCMBs.” Another could be “Thus in, the co-polycondensation of HCTP and moderate ratio of CPE was a good method to generate high-quality MCMBs.” Yet another such unusual phase appears on the abstract could be “What’s more, …” , and is incorrect as far scientific usages are concerned.
2- While the introduction is long, the objective of the study is not clarified.
3- Authors wrote “Coal pyrolytic extracts (CPE) was a kind of aromatic hydrocarbons, which was obtained from low rank coal (such as long-flame coal and brown coal). Theoretically, CPE has a higher reaction reactivity to generate mesophase” . However, no reference is provided to support the claim made of the last sentence.
4- The FTIR spectra in Fig. shows a peak feature in which the authors of this have assigned to the R-CH2-. It is probably misleading since R is the remainder part of the molecular domain and the vibrational mode cannot be assigned to such a fragment. The vibration should be more localized. There is no background reference provided in support of their claim and for comparison. The FTIR spectra needs an elaborative discussion.
5- The conclusion section should be rewritten with transparency, and with no grammatical errors.
Author Response
We thank the reviewer 3 for the decent review comments that lead to improvements of the revised manuscript.Please see below for responses and revisions made based on the reviewer’s comments.

Round 2
Reviewer 3 Report
This work may be considered for possible publications since authors have revised their paper based on my comments and suggestions. I have no other suggestions for improvement.